# Impact and Effectiveness of Legislative Smoking Bans and Anti-Tobacco Media Campaigns in Reducing Smoking among Women in the US: A Systematic Review and Meta-Analysis

**DOI:** 10.3390/healthcare8010020

**Published:** 2020-01-16

**Authors:** Yelena Bird, Ladan Kashaniamin, Chijioke Nwankwo, John Moraros

**Affiliations:** 1Director iCAN Research Group, Brandon, MB R7A 0V6, Canada; yelenabird@gmail.com; 2Department of Community Health and Epidemiology, University of Saskatchewan, Saskatoon, SK S7N 2Z4, Canada; ladan.kashani@usask.ca; 3School of Public Health, University of Saskatchewan, Saskatoon, SK S7N 2Z4, Canada; ckn313@mail.usask.ca; 4Faculty of Health Studies, Brandon University, Brandon, MB R7A 6A9, Canada

**Keywords:** smoking, women, legislative smoking bans, anti-tobacco media campaigns

## Abstract

Background: The purpose of this study is to systematically review the literature addressing the effectiveness of legislative smoking bans and anti-tobacco media campaigns in reducing smoking among women. Methods: MEDLINE, PubMed, CINAHL, and ABI/INFORM were searched for studies published from 2005 onwards. Meta-analysis was conducted using a random effects model and subgroup analysis on pre-selected characteristics. Results: In total, 652 articles were identified, and five studies satisfied the inclusion criteria. The studies varied from school-based to workplace settings and had a total of 800,573 women participants, aged 12 to 64 years old. Three studies used legislative bans, one study used anti-tobacco campaigns and another one used both as their intervention. The overall pooled effect of the five studies yielded an odds ratio (OR) = 1.137 (C.I. = 0.976–1.298 and I^2^ = 85.6%). Subgroup analysis by intervention revealed a significant pooled estimate for studies using legislative smoking bans OR = 1.280 (C.I. = 1.172–1.389 and I^2^ = 0%). Conclusion: Legislative smoking bans were found to be associated with a reduction in the smoking rates among women compared to anti-tobacco media campaigns. Further research in this area is needed.

## 1. Introduction

The first report of the United States (US) Surgeon General’s Advisory Committee on Smoking found that cigarette smoking is a probable cause of lung cancer and poses a serious risk of death and disease for women [1]. More than 50 years later, smoking is still the leading cause of premature death among women in the US and across the world [2,3]. Despite increased awareness of the harm caused by cigarette smoking, the effectiveness of global tobacco control initiatives has been questionable and the gains modest. The World Health Organization (WHO) predicts that cigarette smoking will continue to kill approximately eight million people a year, resulting in more than one billion deaths over the course of the 21st century [4].

Men continue to have higher smoking rates than women in the US and across the world but the gap has steadily decreased over the last couple of decades [5,6]. The narrowing of the gap suggests that women now share a much larger burden of smoking-related diseases, morbidities and mortalities than ever before. For instance, between 1960 and 1990, death rates from lung cancer among US women increased by more than 500% [7]. Starting in the late 1980s, lung cancer surpassed breast cancer to become the leading cause of cancer death among women in the US [8]. Studies have shown the risk for chronic diseases and dying due to smoking to be considerably higher in women than men over the past 50 years [9,10,11].

Tobacco companies have increasingly used a gendered specific approach in their marketing campaigns to effectively target women. Many studies have shown that advertising campaigns by the tobacco industry seek to connect smoking to desirable women behaviours and attributes [12,13,14]. Behaviours are linked to the importance and value of smoking to women in creating fun-loving environments, strong social relationships, positive body image, weight control, independence, social status, sexual desirability and self-relaxation/medication [12]. Attributes such as cigarette size (i.e., long and slim), packaging (i.e., glitzy and sexy), and taste (i.e., light and flavourful) have been changed and designed to attract women. For instance, brands such as Vogue, Silk-Cut, and Virginia Slims have introduced attractive packaging styles like purse packs and a number of limited edition cigarette packs that have been heavily promoted by famous women celebrities and even fashion designers [14].

It is ironic that tobacco companies have linked smoking to women’s independence/social status and well-being and yet cigarette smoking has had the opposite effect on their economic empowerment and physical health. Without empowerment and health, women cannot achieve equality and certainly cannot prosper. Research has shown that girls and women who smoke are more likely to be socioeconomically disadvantaged and/or marginalized [15,16,17]. Therefore, women are a top priority population for tobacco control and prevention efforts. The Framework Convention for Tobacco Control (FCTC) led to a multi-national treaty to help combat the global scourge of tobacco epidemic among a number of vulnerable populations including women [18]. The FCTC identified legislative bans and anti-tobacco media campaigns as important levers to help reduce smoking rates among women [18].

A systematic review by Hoffman et al. found that legislative bans and anti-tobacco media campaigns are effective tools in reducing smoking rates among countries that ratified the FCTC treaty [19]. Additionally, Bala et al. and De Kleijen et al. concluded that mass media campaigns can be effective strategies in smoking reduction and cessation efforts among adults [20,21]. However, there is a significant gap in the literature regarding the systematic assessment of this important topic, specifically among women. Therefore, the purpose of this study is to conduct a systematic review of the literature for quantitative evidence that determines the impact and effectiveness of legislative smoking bans and anti-tobacco media campaigns in reducing smoking among women in the US.

## 2. Methods

### 2.1. Selection of Studies

Identified studies were screened for eligibility by two reviewers. Articles were considered eligible for inclusion in the present study if they: (1) evaluated the effects of legislative smoking bans and/or anti-tobacco media campaigns among populations that included women 15 years old or older; (2) evaluated smoking status before and after the establishment of legislative smoking bans or anti-smoking media campaigns; (3) had a comparison group included in the study; (4) reported quantitative outcome measures specifically for women; and (5) were published in the English language in peer-reviewed journals since 2005, and available in full text.

### 2.2. Search Strategy

Search terms related to legislative smoking bans and anti-tobacco media campaigns were used to search four online databases including: (1) Medline; (2) PubMed; (3) CINAHL; and (4) ABI/INFORM. A grey literature search was also conducted on Google and on ProQuest Dissertations & Thesis Global databases. The references of relevant articles were also carefully reviewed to identify possibly related studies. Search results were imported to separate Excel spreadsheets by using reference management software (Zotero, Corporation for Digital Scholarship, Vienna, Virginia, USA) and duplicate articles were removed.

### 2.3. Data Extraction and Analysis

Using Excel spreadsheets, characteristics of selected studies were extracted including author, publication year, type of study, number of women participants, type of intervention and effect estimates. Crude odds ratios and 95% confidence intervals were computed using the online MedCalc tool for studies that did not provide them but had cross-tabulated data [22]. Meta-analysis was conducted using a random effects model. The random-effects model was used to determine the pooled mean effect size because it enables comparisons between the statistical results arising from the different samples and methodology (measuring methods and units) found among the selected studies [23]. The primary outcome measure was the odds ratio (OR). OR calculations relied on study participant responses based on their smoking habits before and after establishment of legislative smoking bans or anti-smoking media campaigns. ORs and 95% confidence intervals (CIs) were either extracted from the articles or calculated by the authors using the quantitative data provided in the studies.

Statistical analysis for heterogeneity was assessed using Higgins I-squared [24] and further explored with the use of subgroup analysis on predetermined characteristics such as the study design, type of intervention and type of outcome assessed. The robustness of the findings was assessed by determining the influence of each individual study on the overall pooled estimate using Tobias’ method [25]. Publication bias was ascertained using a funnel plot and Egger’s test. All analyses were carried out using Stata/IC version 13.1, College Station, TX: StataCorp LP, College Station, Texas, USA.

## 3. Results

### 3.1. Article Identification

In total, 652 articles were identified (636 from a database search, eight from the grey literature and eight using a snowball search). After removing duplicates, 640 articles underwent a two-step screening process. The first step included a review of all titles and abstracts for relevance. Following this step, 575 studies were excluded. The second step included a careful review of the remaining 65 full text articles. Following this step, only five studies [26,27,28,29,30] met the eligibility criteria and were included in the meta-analysis. The summary of our study selection is shown in a PRISMA diagram (Figure 1).

### 3.2. Study Characteristics

The total number of women participants was 800,573. The age of the women ranged from 12 to 64 years old. All studies were based in the US. The studies varied from school-based [29] to workplace [28] settings. Out of the five studies included, only one [27] used a high-quality study design (i.e., quasi-experimental) with control groups. The other four studies [26,28,29,30] used a lower quality experimental design (i.e., cross-sectional) without control groups. Among the five studies eligible for inclusion, three [27,28,30] used legislative bans, one used anti-tobacco campaigns [29] and another one used both [26] as their intervention. The study characteristics are presented in Table 1.

### 3.3. Pooled Analysis

The overall pooled effect size (ES) of the five studies yielded an OR = 1.137 (C.I. = 0.976–1.298 and I^2^ = 85.6%). Subgroup analysis by study design revealed a significant pooled estimate OR = 1.260 (C.I. = 1.130–1.400) for the quasi-experimental study [27] and a non-significant pooled estimate OR = 1.096 (C.I. = 0.931–1.260) for the cross-sectional studies [26,28,29,30]. Subgroup analysis by intervention revealed a significant pooled estimate OR = 1.280 (C.I. = 1.172–1.389 and I^2^ = 0%) for studies using legislative smoking bans [26,27,28,30] and a non-significant pooled estimate OR = 1.137 (C.I. = 0.976–1.298 and I^2^ = 87.5%) for studies using anti-tobacco media campaigns [26,29] (Figure 2).

### 3.4. Risk of Bias

All five studies were reviewed for risk of bias by using a modified Newcastle Ottawa Scale (NOS) [26,27,28,29,30]. This NOS scale includes three components and eight items: (1) selection of study groups (four items); (2) comparability of the groups (one item); and the (3) ascertainment of the outcomes of interest (three items) [31]. The quality of each study was determined by assigning it to one of three subgroups: (1) good (≥two stars for selection of study groups, one star for comparability of the groups, and three stars for ascertainment of the outcomes of interest components); (2) fair (one star for selection of study groups and two stars for ascertainment of the outcomes of interest components); or (3) poor (0 stars for selection of study groups, 0 stars for comparability of the groups, and ≤one star for ascertainment of the outcomes of interest components). Risk of bias was designated as: low, if there was good quality in all components; unclear/moderate, if there was fair quality in one or more components without poor quality in any components; or high, if there was poor quality in any one of the components. The four cross-sectional studies were found to have an overall moderate risk of bias, whereas the quasi-experimental study had a low risk of bias (Table 2).

## 4. Discussion

Overall, we found that the odds of smoking, though not statistically significant, were reduced by 14%. However, with a stratified pooled analysis by the type of intervention, we found that the odds of achieving smoking reduction among women with the implementation of legislative smoking bans were significantly higher by 28%. Similarly, several studies in the scientific literature corroborate our findings [19,32,33,34]. It is important to note that much of the global progress made in reducing the prevalence of smoking can be specifically attributed to the efficacy of legislative smoking bans [35,36,37]. Smoke-free policies banning smoking in public places and workplaces are known to be the most effective measures. Such policies are shown to help denormalize tobacco use [38], reduce smoking prevalence [39], limit exposure to smoke [40] and mitigate negative health outcomes [41].

Our subgroup analysis found that anti-tobacco media campaigns had no statistically significant effect on smoking among women in the US. The pooled odds of smoking due to the implementation of anti-tobacco media campaign among women increased by 1%. There are several plausible explanations for this finding. We posit this may be a reflection of the broad and non-gender-specific messaging of the majority of anti-tobacco campaigns. Additionally, it has been suggested that the effectiveness of media campaigns may be lessened among women because they watch fewer hours of television, when compared to men, and are, therefore, less likely to be exposed to the televised anti-smoking messages [29]. Our findings contradict the evidence reported by several studies, which show anti-tobacco media campaigns to be a useful tool in the reduction of smoking rates [19,37,42,43,44]. However, it is important to note that the reduction rates reported in the literature were obtained from generalized populations and not specifically for females.

Despite increased global awareness and numerous interventions on this important public health front, smoking rates among women continue to increase dramatically [45,46,47]. This development is concerning and may be in part attributed to changes in the marketing approach employed by the tobacco industry, as a greater focus is now placed on the use of new social media platforms that lack strict regulatory controls [48,49]. Additionally, the tobacco industry expertly uses various gender-based advertising techniques to glamorize smoking in pop culture, as evidenced in many popular movies, music videos and fashion shows [50,51]. These venues are used as social cues to depict female characters as cool, independent, adventurous and edgy and therefore, strongly appeal to a wide range of young females [52]. A recent report found that young people are exposed to an astounding 14.9 billion tobacco impressions in youth related films annually and that the overall number of tobacco incidents within US movies has increased by 72% from 2010 to 2018 [45]. This is an important development as the US Surgeon General found a causal link between exposure to these types of images and smoking initiation, especially among young women [53].

### 4.1. Strengths and Limitations

The present study is one of a few to examine the impact and effectiveness of policy measures (i.e., legislative smoking bans and anti-tobacco media campaigns) on smoking reduction specifically among women in the US. By pooling the various effect estimates, we were able to increase the sample size and thus, the power of the study to assess the desired effect. Our study provides significant evidence that can be used as a reference point for future research. Despite these notable strengths, our study is not without its limitations. First, there are only a small number of studies that use quantifiable data and a sound research methodology to study this topic, which limited our meta-analysis. Second, there was a marked heterogeneity among the included studies and therefore, the pooled results should be interpreted with some degree of caution. Third, the included studies obtained information through follow up surveys but did not account for loss to follow up and its resultant bias. Finally, the majority of the studies were cross-sectional in nature and thus, reported on associations but cannot infer causation.

### 4.2. Implications for Policy and/or Practice

Successfully thwarting and/or reversing increases in tobacco use among women will lead to improved quality of life, positive health outcomes and major disease prevention opportunities. Our findings provide significant implications for interventions aimed at reducing smoking among women in the US. Legislative smoking bans were found to be associated with a reduction in the smoking rates among women, while anti-tobacco media campaigns did not. Legislative smoking bans need to be promoted, enforced and further strengthened. Anti-tobacco media campaigns need to be thoughtfully reviewed and revised and specifically tailored so as to effectively counter the tobacco industry’s targeting of women and expose its deliberate efforts to link smoking with women’s issues of independence, rights, status and progress in society [54].

## 5. Conclusions

The complex and critical connection between smoking and women’s health needs to be widely acknowledged and fully elucidated, along with a gendered-based analysis of tobacco use, advertising and legislation. This meta-analysis sought to determine the impact and effectiveness of counter tobacco marketing and legislative smoking bans in the reduction of smoking rates among women. This topic requires urgent attention, comprehensive policies and further high-quality research (e.g., using control groups for comparison analysis, pre- and post-ban data and robust biochemically measured outcomes), with large sample sizes and longer follow up periods (six months or longer) to determine the most effective strategies for implementation and enforcement of smoking bans to prevent and/or reduce the extent of the tobacco epidemic among women in the US and across the world.

## Figures and Tables

**Figure 1 healthcare-08-00020-f001:**
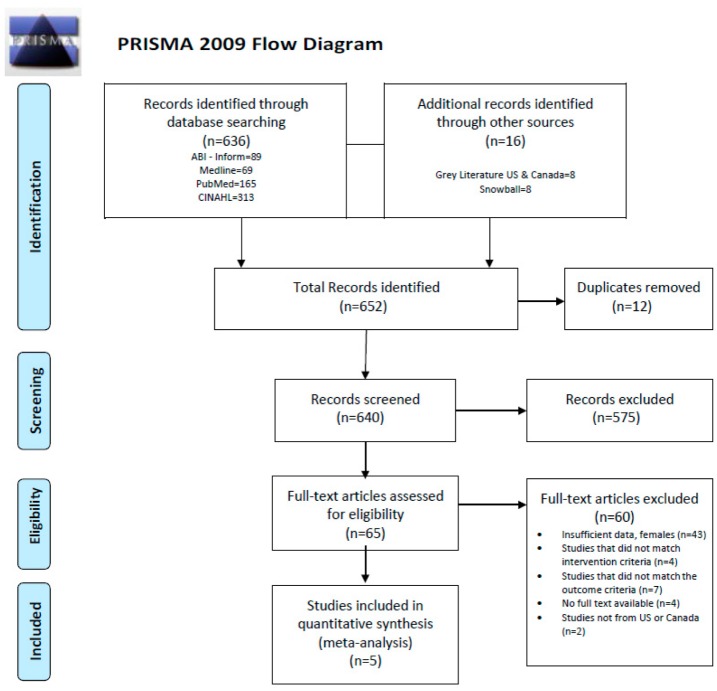
PRISMA diagram, study selection process. Preferred Reporting Items for Systematic Reviews and Meta-Analyses (PRISMA) flow diagram.

**Figure 2 healthcare-08-00020-f002:**
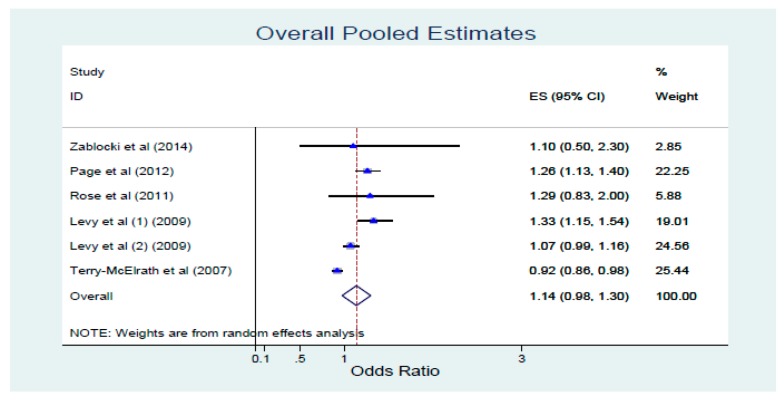
Overall pooled estimates.

**Table 1 healthcare-08-00020-t001:** Study Characteristics.

Year	Author	Purpose	Study Type	Data Source	Number of Females	Age Range of Females (years)	Type of Intervention	Type of Outcome	OR (CI)	Strengths	Limitations
2014	Zablocki et al.	To assess the association of smoking ban policies with smoking reduction and quit attempts among California smokers.	Cross sectional	2011 California longitudinal smokers survey	934	≥18	Home smoking ban, work place smoking ban, perceived city/community smoking ban	Smoking prevalence	OR: 1.1 (0.5–2.3)	Participants are randomly selected, first study to examine the association of perceived city/town smoking bans at outdoor locations with smoking behaviors.	Intervention & outcome data were assessed using self-reported data. Only 50% of the sample participated in the follow-up.
2012	Page et al.	To examine the effect of a citywide smoking ban in comparison to a municipality with no smoking ban in Colorado on maternal smoking outcomes and subsequent fetal birth outcomes.	Natural experiment	State of Colorado Department of Health, Colorado Birth Registry, and the Infant Mortality Registry data	19,769	All ages were included.	Legislative smoking ban	Smoking prevalence	OR: 1.26 (1.13–1.40)	First evidence in regard to improvement of fetal outcomes and preterm birth as a result of smoking ban in the United States. Including a comparison group with same demographics in the study.	Self-reported data is used, Mothers’ exposure to second-hand smoke was not measured directly. Paternal smoking history was not included in the data to estimate SHS. Maternal self-report is probably under-reported due to social stigma related to smoking during pregnancy. Mothers reported lifetime smoking, not in the time period close to the pregnancy.
2011	Rose et al.	To assess the prevalence of work place and home smoking bans and their associations with intention to quit, quit attempts, and 3-month sustained abstinence among employed females.	Cross sectional	Cross-sectional data from the 2006/2007 Tobacco Use Supplement to the Current Population Survey	7610	18–64	Home and work place smoking bans	Smoking prevalence	AOR: 1.29 (0.83–2.00)	First study to examine the association of full smoking bans (at home and work place) with smoking behaviors among employed female smokers. Effect of complete work and home ban was analysed in addition to their separate effects.	Employed indoor females were included in this study. Therefore these data may not be generalizable to all females. The data reported are cross-sectional and do not allow for causal associations. Self-reported data is used. Detailed information such as coworkers and spouse smoking and quitting were not collected in the dataset which they may be influential on smoking behaviour.
2009	Levy et al.	To examine the association between tobacco control policies (clean air laws and media campaigns) with smoking prevalence.	Cross sectional	Tobacco Use Supplement to the Current Population Survey 1992–2002	Total sample: 707,720 (Number of females not mentioned)	≥18	Antismoking policies, Anti-smoking media campaigns	Smoking prevalence	OR (Clean air): 1.33 (1.15–1.54) OR (Media): 1.07 (0.99–1.16)	Examining the effect of different tobacco control policies on smoking prevalence. A dataset related to a large population was used. Age and gender variations in addition to variations over time were considered in the study.	Different forms of policies that may have different effects, were included to the policy measure. Socio-economic factors were not considered in the study.
2007	Terry-McElrath et al.	To examine the association between anti-tobacco advertising and smoking related outcomes with respect to gender and race/ethnicity	Cross sectional	8th, 10th, and 12th grades student data in 1999–2003 collected by Monitoring the Future study	64,840	14–18	Anti-smoking media campaigns	Smoking prevalence	OR: 0.92 (0.86–0.98)	First study to examine the association between exposure to anti-tobacco advertising and smoking outcomes in 8th, 10th, 12th grades students. Comparison among males and females and among different racial/ethnic groups was performed.	Hispanics were included in the study population. However, Spanish-language TV channels were not included.

**Table 2 healthcare-08-00020-t002:** Risk of bias assessment using modified Newcastle Ottawa Scales (NOSs).

NOS. Cross Sectional		Selection				Comparability			Outcome			Risk of Bias
Year	Author	Representativeness	Ascertainment of Exposure	Outcome at start	Rating	Controls for Gender	Controls for covariates	Rating	Assessment of outcome	Completeness of outcome	Rating	
2014	Zablocki et al.	1	0	1	Good	1	1	Good	0	1	Fair	Moderate
2011	Rose A et al.	1	1	1	Good	1	1	Good	0	1	Fair	Moderate
2009	Levy et al.	1	1	1	Good	1	1	Good	0	1	Fair	Moderate
2007	Terry-McElrath et al.	1	1	1	Good	1	1	Good	0	1	Fair	Moderate
NOS Quasi Experimental		Selection				Comparability			Outcome			Risk of bias
Year	Author	Representativeness	Ascertainment of Exposure	Outcome at start	Rating	Controls for Gender	Controls for covariate	Rating	Assessment of outcome	Completeness of outcome	Rating	
2012	Page et al.	1	1	1	Good	1	1	Good	0	1	Good	Low

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
