# Peer review of "Impact and Effectiveness of Legislative Smoking Bans and Anti-Tobacco Media Campaigns in Reducing Smoking among Women in the US: A Systematic Review and Meta-Analysis"

_healthcare, 2020, doi:10.3390/healthcare8010020_

Round 1

Reviewer 1 Report

This manuscript conducts a meta analysis in order to assess the impact of different strategies of curbing tobacco use in women—legislative bans and media campaigns. The authors conclude that bans were more likely than media campaigns to reduce smoking rates in women. I offer a few suggestions for improvement below.

There are several tables and figures in the body of the desk that are lacking in discussion and explanation. The authors should spend more time describing these graphics and their contribution to the manuscript.

The authors discuss why media campaigns might not decrease smoking in women, but the discussion could also benefit from a discussion of why smoking bans were more successful.

The focus of the manuscript is on media campaigns and legislative bans; however, the authors conclude that future research is needed. In the conclusion, it may be useful for the authors to specifically mention where they believe future research should focus.

Author Response

OVERALL

We are thankful to the editorial team and the Reviewers for their careful review of our manuscript, insightful recommendations for revisions and helpful feedback. It is thanks to their valuable input that our revised manuscript represents a considerably stronger, better and higher quality submission.

REVIEWER 1

Comments and Suggestions for Authors

Comment 1:

This manuscript conducts a meta-analysis in order to assess the impact of different strategies of curbing tobacco use in women—legislative bans and media campaigns. The authors conclude that bans were more likely than media campaigns to reduce smoking rates in women. I offer a few suggestions for improvement below.

Response 1:

Thank you for taking the time to critically review our manuscript and for kindly providing us with your helpful suggestions.

Comment 2:

There are several tables and figures in the body of the desk that are lacking in discussion and explanation. The authors should spend more time describing these graphics and their contribution to the manuscript.

Response 2:

Thank you for your comment. Please note that the number of Figures in our manuscript has been reduced as recommended by another Reviewer. Additionally, in the revised draft, all Tables and Figures are clearly identified, mentioned and described in the text of the manuscript [Page 6; Section: Risk of Bias; First paragraph].

Comment 3:

The authors discuss why media campaigns might not decrease smoking in women, but the discussion could also benefit from a discussion of why smoking bans were more successful.

Response 3:

Thank you for your astute observation. We agree that our discussion can benefit from a section of why smoking bans are known to be more successful and provide context for their positive impact. We have incorporated such a section in our discussion [Page 7, Section: Discussion; First paragraph, last sentence].

Comment 4:

The focus of the manuscript is on media campaigns and legislative bans; however, the authors conclude that future research is needed. In the conclusion, it may be useful for the authors to specifically mention where they believe future research should focus.

Response 4:

Thank you for your insightful suggestion. We agree that it will be useful to specifically mention where we believe future research should focus. Therefore, we have identified possible future research areas of interest in our conclusion and specifically, mention several areas that need to be examine in order to help us determine the most effective strategies in the implementation and enforcement of smoking bans [Page 8, Section: Conclusion; First paragraph, last sentence].

Reviewer 2 Report

A major problem with the paper is that both the title of the paper and the abstract talk about “effect” or “effectiveness”, but give the reader no clues to what this effectiveness relates to.  Presumably, it is effectiveness in reducing the prevalence of smoking, but this should be clearly stated.  Perhaps I have misunderstood, but the two Figures on page 5 have two meta-analyses, one headed “overall pooled estimates” and one headed “pooled estimates by outcome type” where the results are exactly the same!  As all the estimates relate to smoking prevalence, it is unclear to me what other outcome types the authors might have considered, or why both halves of Figure 1 are needed. 

Another fundamental point is that no explanation seems to be given on how the effect estimate was derived, which is extremely important to know.  Related to this is the fact that Table 1, which has lost its left hand and right hand sides, is so small that it is very difficult to read.  There appear to be two main ways to assess the effect of a ban.  One is to compare changes in prevalence over the same time period in a “test” area where the ban was introduced during the ban, and a “control” area where it was not.  The other is to compare the prevalence following the ban with that predicted by fitting a trend to data over a series of years before the ban was introduced.  In some of the studies it is not clear how either approach applies.  For instance, how can one estimate the effect of a ban from a cross-sectional study with data collected at one time point, as is apparently the situation with the first study in Table 1?  It is much more important to give clear details of how the outcome was estimated in the different studies than to give less important information such as funnel plots.

Other points are as follows:

In the abstract it is stated that there were five studies, of which four used legislative smoking bans and two used anti-tobacco media campaigns.  As 5 is not equal to 4 + 2, it would have been better to say three used legislative bans, one anti-tobacco campaigns and one both.  This point arises later in the paper as well.

The abstract and the paper should give the result of a formal test of significance for the difference between results for the two different types of ban.

In Figure 1, the bottom right hand box states n = 60, but the individual numbers only add to 58.

Does one really need four meta-analysis figures?  I would have thought it quite adequate to only give one, to state the outcome was the same for all the studies, and in the text give the combined estimates separated by study type and intervention type - and a test of whether results varied by these factors.

As noted Table 1 is too small, and probably should be landscape or separated into sections, each with only a few columns.

In Table 2 Newcastle is one word, not two.

ES is unexplained in the meta-analysis figures.

In the funnel plot, the over printing could be easily avoided by giving Terry-McGrath et al left of its dot.

In the discussion, it is stated that “several studies in the scientific literature corroborate our findings” citing four references.  Are these previous meta-analyses?  Did they use any information not used in the current paper?

In paragraph 2 of the discussion it is stated that “the pooled odds of smoking due to the implementation of anti-tobacco media campaign among women decreased by only 1%”.  Should this not be increased, as the OR for smoking reduction was 0.99?  On this point is should be made clear everywhere what the OR (ES) relates to.

I note that in the limitations it is correctly noted that as the majority of the studies were cross-sectional, one cannot infer causation.  But the wording in many places ignores this – e.g. the conclusion sentence in the abstract; “was associated with a reduction” not “significantly reduce”.

Author Response

OVERALL

We are thankful to the editorial team and the Reviewers for their careful review of our manuscript, insightful recommendations for revisions and helpful feedback. It is thanks to their valuable input that our revised manuscript represents a considerably stronger, better and higher quality submission.

REVIEWER 2

Comments and Suggestions for Authors

Comment 1:

A major problem with the paper is that both the title of the paper and the abstract talk about “effect” or “effectiveness”, but give the reader no clues to what this effectiveness relates to.  Presumably, it is effectiveness in reducing the prevalence of smoking, but this should be clearly stated. 

Response 1:

Thank you for your helpful comment. Please note that we have incorporated your helpful suggestion in several relevant sections of our revised manuscript as appropriate [Title; Abstract; and Text].

Comment 2:

Perhaps I have misunderstood, but the two Figures on page 5 have two meta-analyses, one headed “overall pooled estimates” and one headed “pooled estimates by outcome type” where the results are exactly the same!  As all the estimates relate to smoking prevalence, it is unclear to me what other outcome types the authors might have considered, or why both halves of Figure 1 are needed. 

Response 2:

Thank you for your astute observation. You are correct the additional Figures are not needed. Therefore, we have kept Figure 1 and removed all other meta-analyses figures. This was done in an effort to be concise with the presentation of our information and to avoid redundancy and possible confusion.

Comment 3:

Another fundamental point is that no explanation seems to be given on how the effect estimate was derived, which is extremely important to know.  Related to this is the fact that Table 1, which has lost its left hand and right hand sides, is so small that it is very difficult to read.  There appear to be two main ways to assess the effect of a ban.  One is to compare changes in prevalence over the same time period in a “test” area where the ban was introduced during the ban, and a “control” area where it was not.  The other is to compare the prevalence following the ban with that predicted by fitting a trend to data over a series of years before the ban was introduced.  In some of the studies it is not clear how either approach applies.  For instance, how can one estimate the effect of a ban from a cross-sectional study with data collected at one time point, as is apparently the situation with the first study in Table 1?  It is much more important to give clear details of how the outcome was estimated in the different studies than to give less important information such as funnel plots.

Response 3:

Thank you for your insightful comment. A more fulsome discussion is now provided and helps add clarity to the importance of the size effect and how it was derived and why it was used in our study [Page 3, Section: Methods; Subsection: Data Extraction and Analysis; First paragraph, last two sentences with an additional reference].

Comment 4:

Other points are as follows:

In the abstract it is stated that there were five studies, of which four used legislative smoking bans and two used anti-tobacco media campaigns.  As 5 is not equal to 4 + 2, it would have been better to say three used legislative bans, one anti-tobacco campaigns and one both.  This point arises later in the paper as well.

Response 4:

This is an excellent point. The language has been corrected accordingly both in the Abstract and in the text of the manuscript (Please see Abstract; Page 1; Section: Results and Text; Page 3; Section: Results; Subsection: Study Characteristics; First paragraph, last two sentences).

Comment 5:

The abstract and the paper should give the result of a formal test of significance for the difference between results for the two different types of ban.

Response 5:

Thank you for your comment. Please note that both in the Abstract and the text of our manuscript, we clearly present the result of a formal test of significance for the difference between results for the two different types of ban (Please see Abstract “Sub-group analysis by intervention revealed a significant pooled estimate for studies using legislative smoking bans OR= 1.280 (C.I. = 1.172-1.389 and I2 =0%).”)

Comment 6:

In Figure 1, the bottom right hand box states n = 60, but the individual numbers only add to 58.

Response 6:

This is an excellent observation. Thank you. It has now been corrected.

Comment 7:

Does one really need four meta-analysis figures?  I would have thought it quite adequate to only give one, to state the outcome was the same for all the studies, and in the text give the combined estimates separated by study type and intervention type - and a test of whether results varied by these factors.

Response 7:

You are absolutely correct. There is no need for four meta-analysis figures. As suggested, we have retained only one figure (Figure 2. Overall Pooled Estimates) and provided the text to accompany the interpretation of its key findings.

Comment 8:

As noted Table 1 is too small, and probably should be landscape or separated into sections, each with only a few columns.

Response 8:

Please accept our apologies and note that we originally submitted all of our Tables and Figures in easy to read formats and sizes as a separate document. In particular, Table 1 was submitted as a Word document and in landscape format. The format was changed and incorporated into the manuscript by the editorial staff, presumably to facilitate review. In so doing, it was converted to portrait and its left and right margins were missing. We have attempted to assist the editorial staff by reintroducing the full table (albeit it is quite small and potentially difficult to read) and once again we submitted it as a separate Word document in landscape format and using a larger font size.

Comment 9:

In Table 2 Newcastle is one word, not two.

Response 9:

Thank you. Please note the correction is made.

Comment 10:

ES is unexplained in the meta-analysis figures.

Response 10:

Thank you. ES is the “Effect Size” and it is now explained in the text as it relates to the meta-analysis figure (Figure 2).

Comment 11:

In the funnel plot, the over printing could be easily avoided by giving Terry-McGrath et al left of its dot.

Response 11:

The funnel plot has been removed as previously recommended by the Reviewer (Comment 3 above).

Comment 12:

In the discussion, it is stated that “several studies in the scientific literature corroborate our findings” citing four references.  Are these previous meta-analyses?  Did they use any information not used in the current paper?

Response 12:

This is an excellent question. Please note that of the four studies used to corroborate our findings only one was a meta-analysis (Hoffman, S.J.; Tan, C. Overview of systematic reviews on the health-related effects of government tobacco control policies. BMC Public Health 2015, 15). The aim of the Hoffman and Tan systematic review was to examine the global research evidence about the likely general health effects of different government tobacco control policies. This is quite different in scope, aim and purpose from our own study and therefore, Hoffman and Tan used information and research papers that are different from the ones used in the current paper.

Comment 13:

In paragraph 2 of the discussion it is stated that “the pooled odds of smoking due to the implementation of anti-tobacco media campaign among women decreased by only 1%”.  Should this not be increased, as the OR for smoking reduction was 0.99?  On this point is should be made clear everywhere what the OR (ES) relates to.

Response 13:

Thank you for your astute observation. It has been corrected to now read “increased”.

Comment 14:

I note that in the limitations it is correctly noted that as the majority of the studies were cross-sectional, one cannot infer causation.  But the wording in many places ignores this – e.g. the conclusion sentence in the abstract; “was associated with a reduction” not “significantly reduce”.

Response 14:

This is another excellent observation. Please note that the language has been appropriately revised to accurately read as recommended “associated with a reduction”.

Round 2

Reviewer 2 Report

Although it is significantly improved, I would prefer that a few points are taken into account before it is published.

The first point is the most important - others are less so, some being correcting minor errors.

1. It is still not clear to me how the ORs are calculated. When estimating the effect of smoking bans one can use various different approaches.

a) Comparing smoking prevalences at one point in time between areas where a ban has or has not been introduced (or between areas with different levels of ban strictness)

b) Comparing changes in smoking prevalence in an area between time points before and after a ban was introduced.

c) Comparing changes in smoking prevalence as in b) with changes occurring between the same time points in areas where a ban was not introduced.

The selection of studies section (2.1) is somewhat unclear about this.

Statements 2) and 3) suggest to me that the studies had to use approach c), as they had to determine prevalence at two time points and had to have a control group, but this would seem to require determination of smoking prevalence at different time points.

However the studies selected are mainly cross-sectional - did the studies really ask individuals about their smoking habits before and after the ban? If so, this should be made clear.

Linked to this point is Table 1. It is very much more important to make clear to the reader the method of calculating the OR than to give some of the information that is included in that table.

Thus it is just confusing and unhelpful to include under types of outcome things like low birth weight which I assume have nothing at all to do with the estimated OR.

An unadjusted OR is based on a 2x2 breakdown of the data - make it absolutely clear what this is for each study.

2. Another issue is that you have two ORs for the Levy study. Are they really based on independent samples? If not, only one of them should be used in any one meta-analysis.

3. Other points are minor:

a) Section 2.3 In the sentence "The primary outcome measure was odds ratio (OR)" insert "the" before "odds".

b) Section 3.2 In the final sentence inset "The" before "study characteristics".

c) There is inconsistency in Figure 1 as to whether there is a space before the final bracket in statements such as "(n=5 )". It would be better without the spaces.

d) In Table 1 the column "OR, CI" should make it clear which type of outcome the OR relates to.

e) In Table 2 it is "COMPARABILITY" not "COMPARIBILITY".

Author Response

OVERALL

We thank the editorial team and the Reviewer 2 for their encouragement and second round of helpful feedback.

REVIEWER 2

Comment:

Although it is significantly improved, I would prefer that a few points are taken into account before it is published.

Response:

Thank you for helping us improve our draft and bringing these additional key points to our attention. We have made every effort to address them in our second revision of our manuscript as explained below.

Comment 1:

The first point is the most important - others are less so, some being correcting minor errors.

It is still not clear to me how the ORs are calculated. When estimating the effect of smoking bans one can use various different approaches. a) Comparing smoking prevalences at one point in time between areas where a ban has or has not been introduced (or between areas with different levels of ban strictness) b) Comparing changes in smoking prevalence in an area between time points before and after a ban was introduced. c) Comparing changes in smoking prevalence as in b) with changes occurring between the same time points in areas where a ban was not introduced.

The selection of studies section (2.1) is somewhat unclear about this.

Statements 2) and 3) suggest to me that the studies had to use approach c), as they had to determine prevalence at two time points and had to have a control group, but this would seem to require determination of smoking prevalence at different time points.

However the studies selected are mainly cross-sectional - did the studies really ask individuals about their smoking habits before and after the ban? If so, this should be made clear.

Linked to this point is Table 1. It is very much more important to make clear to the reader the method of calculating the OR than to give some of the information that is included in that table.

Thus it is just confusing and unhelpful to include under types of outcome things like low birth weight which I assume have nothing at all to do with the estimated OR.

An unadjusted OR is based on a 2x2 breakdown of the data - make it absolutely clear what this is for each study.

Response 1:

We apologize for any confusion and thank you for your comment. As you noted this information is in part presented in our selection criteria (2.1 Selection of Studies, criteria 2 and 3). However, thanks to your helpful suggestion, we have now also introduced language that clearly articulates “OR calculations relied on study participant responses based on their smoking habits before and after establishment of legislative smoking bans or anti-smoking media campaigns.” (2.3. Data Extraction and Analysis)

Relatedly, please note that as per your recommendation, we have streamlined our presentation of Table 1 by removing information from the “Types of Outcomes” column that is unrelated to our study and potentially confusing to our readers.

Comment 2:

Another issue is that you have two ORs for the Levy study. Are they really based on independent samples? If not, only one of them should be used in any one meta-analysis.

Response 2:

Thank you for your excellent point. Please note that you are correct. They are based on independent samples examining the association between two relevant tobacco control policies (one measuring the effect of a clean air policy and the other measuring a media campaign) with smoking prevalence. Therefore, both were used in our meta-analysis.

Comment 3:

Other points are minor: a) Section 2.3 In the sentence "The primary outcome measure was odds ratio (OR)" insert "the" before "odds".

Response:

Thank you. Please note “the” has been inserted before “odds.”

b) Section 3.2 In the final sentence inset "The" before "study characteristics".

Response:

Thank you. Please note “The” has been inserted before “study characteristics.”

c) There is inconsistency in Figure 1 as to whether there is a space before the final bracket in statements such as "(n=5 )". It would be better without the spaces.

Response:

Thank you. We have corrected the inconsistencies “(n=5)”and present all the relevant information without spaces.

d) In Table 1 the column "OR, CI" should make it clear which type of outcome the OR relates to.

Response:

Thank you. We now made it clear that the OR pertains to smoking prevalence by following your recommendation and being consistent in our use of language in the preceding column “Type of outcome.”

e) In Table 2 it is "COMPARABILITY" not "COMPARIBILITY".

Response 3:

Our apologies for the spelling error and we thank you for bringing it to our attention. It is now corrected.